# Preserving Marker Specificity with Lightweight Channel-Independent Representation Learning

**Simon Gutwein**[1,2,5]                                    SIMON.GUTWEIN@CCRI.AT
**Arthur Longuefosse**[3]                                ARTHUR.LONGUEFOSSE@RIKEN.JP
**Jun Seita**[3,4]                                                JUN.SEITA@RIKEN.JP
**Sabine Taschner-Mandl**[1]                            SABINE.TASCHNER@CCRI.AT
**Roxane Licandro**[2,5,6]                          ROXANE.LICANDRO@MEDUNIWIEN.AC.AT

[1] *St. Anna Children's Cancer Research Institute, Vienna, Austria*

[2] *Medical University of Vienna, Biomedical Imaging and Image-guided Therapy, Computational Imaging Research, ELIA Group, Vienna, Austria*

[3] *RIKEN Center for Integrative Medical Sciences, Medical Data Deep Learning Team, Tokyo, Japan*

[4] *RIKEN Center for Interdisciplinary Theoretical and Mathematical Sciences, Tokyo, Japan*

[5] *Medical University of Vienna, Comprehensive Center for AI in Medicine, Vienna, Austria*

[6] *Medical University of Vienna, Christian Doppler Lab for Mathematical Modelling and Simulation of Next-Generation Medical Ultrasound Devices, Vienna, Austria*

## Abstract

Multiplex tissue imaging measures dozens of protein markers per cell, yet most deep learning models still apply early channel fusion, a design choice that erroneously imposes strong shared structure across weakly correlated channels. We present a systematic study of channel fusion as an architectural design choice for self-supervised representation learning. Using a Hodgkin lymphoma CODEX dataset with 145,000 cells and 49 markers, we demonstrate that preserving channel isolation yields substantially stronger representations than early fusion. Notably, a shallow channel-isolated architecture with only 5.5K parameters matches state-of-the-art models. We further demonstrate that channel-isolated embeddings enable interpretable phenotyping via marker attribution. Blind expert validation confirms that the proposed method produces annotations that respect biological constraints better than conventional pipelines (82.1% vs. 54.1% agreement). Our results show that delayed channel fusion can substitute for model scale in multiplex imaging, yielding interpretable representations that align more closely with biological ground truth and enable reliable expert-validated phenotyping. Code is available at `https://github.com/SimonBon/CIM-S`.

**Keywords:** Multiplex Imaging, Representation Learning, Cell Phenotyping

## 1. Introduction

Biological Multiplex Imaging (MI) technologies quantify multiple protein markers at subcellular resolution, providing high-dimensional data on single-cell states and tissue organization (Goltsev et al., 2018; Giesen et al., 2014). Widely used analysis pipelines rely on cell segmentation (Windhager et al., 2023) followed by aggregation of per-cell intensities. This approach propagates segmentation errors into downstream analyses (Bruhns et al., 2025) and restricts representation learning to predefined features. Convolutional Neural Networks (CNNs) offer an alternative by learning directly from raw images; however, standard architectures like ResNet (He et al., 2016) implement early-fusion, linearly combining all input channels in the first layer (Wang et al., 2024).

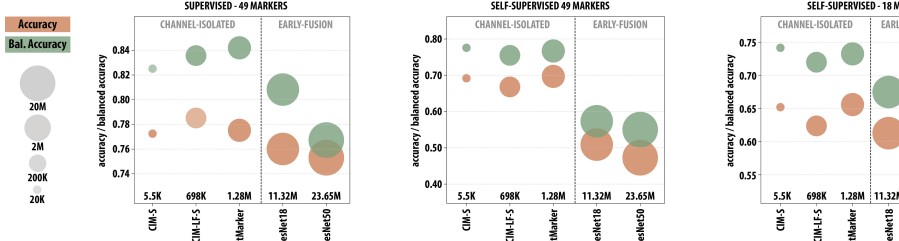

Figure 1: Representation-quality benchmark using self-supervised learning. Bubble size indicates parameter count. **Left:** Evaluation on all 49 markers. **Right:** Evaluation on a reduced 18-marker panel, including comparison with the KRONOS foundation model.

While early fusion is suited for highly correlated RGB images, MI channels represent independent molecular markers. Premature fusion can induce modality collapse (Chaudhuri et al., 2025), where weaker predictive signals are masked and essential markers cease to contribute to learning. We hypothesize that preserving marker independence in early layers is a critical inductive bias for MI. To test this, we systematically evaluate channel-isolated architectures, such as NeXtMarker (Gutwein et al., 2025), against early-fusion baselines and the KRONOS foundation model (Shaban et al., 2025) under self-supervised learning (SSL). We then demonstrate an interpretable, segmentation-free phenotyping pipeline using Layer-wise Relevance Propagation (LRP) (Bach et al., 2015) to automatically extract accurate cell labels.

## 2. Methods and Experiments

We utilize a publicly available CODEX dataset of classical Hodgkin lymphoma (cHL) (Shaban et al., 2024), containing approximately 145,000 cells across 49 protein markers. The baseline dataset includes annotations generated via DeepCell segmentation (Greenwald et al., 2022) and expert curation.

**Proposed Architectures and SSL Benchmark:** To isolate the effect of fusion strategy, we propose a lightweight Channel-Isolated Model (**CIM-S**). CIM-S ($\approx$ 5.5K params) uses grouped convolutions to preserve channel separation; **CIM-LF-S** adds late fusion (LF). We benchmark against ResNets (**ResNet18/50**), which serve as early-fusion baselines. Models were pretrained using the contrastive framework SimCLR (Chen et al., 2020). Subsequently, the encoders were frozen and evaluated via linear probing for cell classification.

**Linear Probing Results:** Early-fusion CNNs struggle to retain marker-specific information (Figure 1), with balanced accuracies of 55–58%. CIM-S achieves 77.6%, slightly outperforming NeXtMarker (76.7%) and the late-fusion variant CIM-LF-S (75.5%). When evaluated on a reduced 18-marker panel, the compact 5.5K-parameter CIM-S slightly outperforms the 21M-parameter KRONOS foundation model (74.2% vs. 73.6%). This demonstrates that architectural inductive bias can effectively substitute for massive model scale.

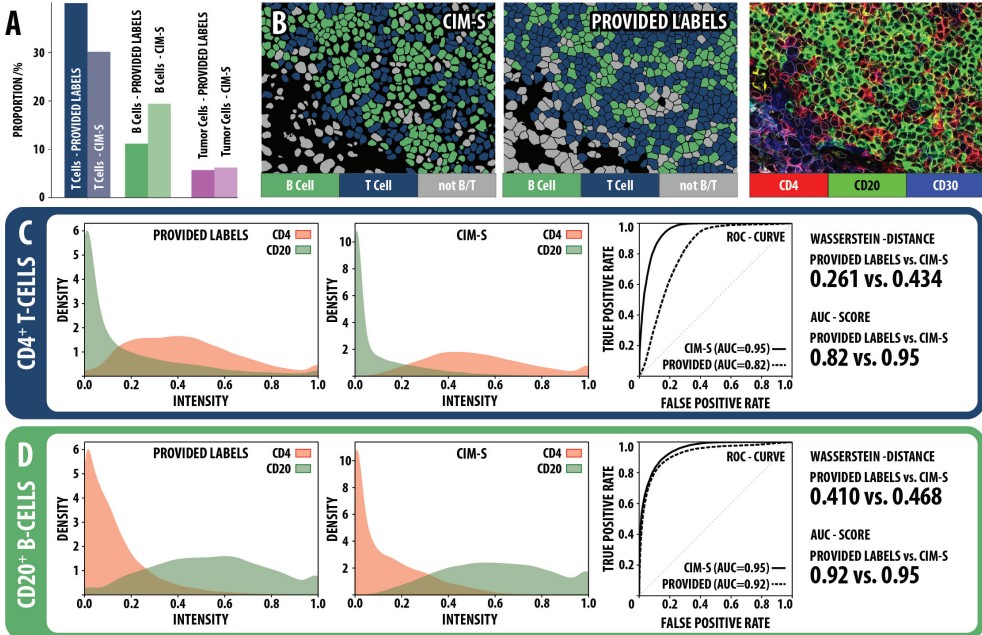

Figure 2: Quantitative comparison between CIM-S labels and provided baseline labels. **A:** Inferred cell-type proportions. **C/D:** CD4/CD20 separability within T-cell and B-cell populations, quantified by Wasserstein distance and ROC/AUC.

**Label-Free Phenotyping:** Frozen CIM-S representations enable segmentation-free phenotyping via LRP: spatial relevance maps are aggregated to channel-wise scores and multiplied with a marker-phenotype matrix (e.g., CD4→T-cells, CD20→B-cells).

Comparing our LRP-derived labels to the provided dataset labels revealed a strong shift: CIM-S identified 72.7% more B cells and 24.8% fewer T cells. To objectively evaluate this, we analyzed CD4 (T-cell) and CD20 (B-cell) signals, which are biologically mutual exclusive. CIM-S+LRP assignments achieved substantially higher CD4/CD20 separability within the T-cell population (WD, AUC in Figure 2) than the baseline provided cell type labels (Fig. 2). Blind evaluation with an external immunology expert on ∼100 conflicting cells (no access to either label set) confirmed CIM-S superiority: 82.1% agreement vs. 54.1% for baseline labels, indicating systematic errors in conventional segmentation-clustering pipelines.

## 3. Conclusion

Our systematic evaluation demonstrates that preserving channel isolation in early layers prevents modality collapse in MI, allowing a 5.5K-parameter model to match a 21M-parameter foundation model. Furthermore, coupling channel-isolated embeddings with LRP enables interpretable, segmentation-free phenotyping. Blind expert validation confirmed that our pipeline successfully corrects systematic artifacts present in established provided labels, overcoming the "gold standard paradox" in digital pathology.

## Acknowledgements

This research was funded by the Austrian Science Fund (FWF): the Emerging Fields grant DART2OS – Devising Advanced TCR-T cells to eradicate OsteoSarcoma (DOI: 10.55776/EFP45) and the Stand-Alone Project MAPMET – Mapping metastatic cancer by multi-modal imaging (DOI: 10.55776/P35841), both led by Sabine Taschner-Mandl (CCRI).

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
