# OpenReview forum: "Preserving Marker Specificity with Lightweight Channel-Independent Representation Learning"
_MIDL.io/2026/Short_Papers — MIDL 2026 - Short Papers Poster_

### Official Review · Reviewer_99Ad · 2026-05-03
**Channel isolation as an inductive bias for deep learning models in multiplex imaging**

**Rating:** 4
**Confidence:** 4

**Review:**

This short paper is well-written and addresses a topic of significant interest to the medical imaging community. The authors make a well-motivated argument that channel isolation is a critical inductive bias for representation learning in multiplex tissue imaging, and the results are strong, with CIM-S consistently and substantially outperforming early-fusion ResNet baselines. The blind expert validation further strengthens the contribution. The main limitation is that the performance advantage is confounded with model size, as reduced overfitting remains an alternative explanation.

**Summary:**

This short paper investigates channel fusion strategy as an architectural design choice for deep learning on multiplex tissue imaging data. The authors propose CIM-S, a lightweight (5.5K parameter) channel-isolated model, and benchmark it against early-fusion ResNets (18/50) and the KRONOS foundation model. CIM-S consistently outperforms ResNet baselines in both supervised and self-supervised settings on the full 49-marker panel, and slightly outperforms KRONOS in the self-supervised setting on a reduced 18-marker panel. Additionally, the authors couple CIM-S with Layer-wise Relevance Propagation (LRP) for segmentation-free cell phenotyping, and validate the resulting labels against an external immunology expert, achieving 82.1% agreement versus 54.1% for conventional pipeline labels.

**Strengths:**

- The paper is clearly written and easy to follow.
- The empirical results are strong, with the proposed model outperforming ResNet baselines and the KRONOS foundation model.
- The blind expert validation is a major strength, providing evidence of biological relevance.
- The LRP-based phenotyping pipeline is elegant and useful, offering interpretability and enabling segmentation-free cell type annotation.

**Weaknesses:**

- Confound between architecture and model size. CIM-S is substantially smaller than the compared models (which is presented as an advantage), but it remains unclear whether the gains are due to channel isolation or model size. Evaluating channel isolation in size-matched architectures would help isolate this effect.
- LRP is a post-hoc interpretability method. Since LRP can be applied to any NN architecture, it is unclear whether the phenotyping improvements stem from channel isolation or from the method itself.
- Experiments are restricted to a single dataset.

**Justification Of Rating:**

This short paper addresses a relevant problem and presents strong results, with clear improvements over early-fusion baselines and competitive performance with a foundation model. The blind expert validation is a notable strength.

The main limitation is the confound between architecture and model size, making it unclear whether gains are due to channel isolation. Additionally, LRP is not specific to the proposed model, and evaluation is limited to a single dataset.

Overall, this work is of value to be discussed as a short paper at MIDL, and I therefore recommend acceptance.

---

### Decision · Program_Chairs · 2026-05-08

Accept (Poster)